# Sporadic outbreaks of crimean-congo haemorrhagic fever in Uganda, July 2018-January 2019

**Bernadette Basuta Mirembe**[1]\*, **Angella Musewa**[1], **Daniel Kadobera**[1], **Esther Kisaakye**[1], **Doreen Birungi**[1], **Daniel Eurien**[1], **Luke Nyakarahuka**[2,3], **Stephen Balinandi**[2,3], **Alex Tumusiime**[2], **Jackson Kyondo**[2], **Sophia Mbula Mulei**[2], **Jimmy Baluku**[2], **Benon Kwesiga**[1], **Steven Ndugwa Kabwama**[1], **Bao-Ping Zhu**[4,5], **Julie R. Harris**[4,5], **Julius Julian Lutwama**[2], **Alex Riolexus Ario**[1,6]

**1** Uganda Public Health Fellowship Program, Ministry of Health, Kampala, Uganda, **2** Uganda Virus Research Institute, Entebbe, Uganda, **3** School of Biosecurity, Biotechnical and Laboratory Sciences, College of Veterinary Medicine, Animal Resources & Biosecurity, Makerere University, Kampala, Uganda, **4** US Centers for Disease Control and Prevention, Kampala, Uganda, **5** Division of Global Health Protection, Center for Global Health, Atlanta, GA, United States of America, **6** Ministry of Health, Kampala, Uganda

\* bagheni@musph.ac.ug

**Data Availability Statement:** All relevant data are within the manuscript.

**Funding:** This project was supported by the President's Emergency Plan for AIDS Relief

## Abstract

### Introduction

Crimean-Congo haemorrhagic fever (CCHF) is a tick-borne, zoonotic viral disease that causes haemorrhagic symptoms. Despite having eight confirmed outbreaks between 2013 and 2017, all within Uganda's 'cattle corridor', no targeted tick control programs exist in Uganda to prevent disease. During a seven-month-period from July 2018-January 2019, the Ministry of Health confirmed multiple independent CCHF outbreaks. We investigated to identify risk factors and recommend interventions to prevent future outbreaks.

### Methods

We defined a confirmed case as sudden onset of fever ($\geq 37.5°C$) with $\geq 4$ of the following signs and symptoms: anorexia, vomiting, diarrhoea, headache, abdominal pain, joint pain, or sudden unexplained bleeding in a resident of the affected districts who tested positive for Crimean-Congo haemorrhagic fever virus (CCHFv) by RT-PCR from 1 July 2018–30 January 2019. We reviewed medical records and performed active case-finding. We conducted a case-control study and compared exposures of case-patients with age-, sex-, and sub-county-matched control-persons (1:4).

### Results

We identified 14 confirmed cases (64% males) with five deaths (case-fatality rate: 36%) from 11 districts in western and central region. Of these, eight (73%) case-patients resided in Uganda's 'cattle corridor'. One outbreak involved two case-patients and the remainder involved one. All case-patients had fever and 93% had unexplained bleeding. Case-patients were aged 6–36 years, with persons aged 20–44 years more affected (AR: 7.2/1,000,000)

(PEPFAR) through the US Centers for Disease Control and Prevention Cooperative Agreement number GH001353–01 through Makerere University School of Public Health to the Uganda Public Health Fellowship Program, Ministry of Health. Its contents are solely the responsibility of the authors and do not necessarily represent the official views of the US Centers for Disease Control and Prevention, the Department of Health and Human Services, Makerere University School of Public Health, or the Ministry of Health. The staff of the funding body provided technical guidance in the design of the study, ethical clearance and collection, analysis, and interpretation of data and in writing the manuscript. Funding for the study was received through facilitation of field visits during investigations and general Fellowship activities not specific to the study.

**Competing interests:** The authors have declared that no competing interests exist.

than persons $\leq$19 years (2.0/1,000,000), p = 0.015. Most (93%) case-patients had contact with livestock $\leq$2 weeks before symptom onset. Twelve (86%) lived <1 km from grazing fields compared with 27 (48%) controls ($OR_{M-H}$ = 18, 95% CI = 3.2-$\infty$) and 10 (71%) of 14 case-patients found ticks attached to their bodies $\leq$2 weeks before symptom onset, compared to 15 (27%) of 56 control-persons ($OR_{M-H}$ = 9.3, 95%CI = 1.9–46).

## Conclusions

CCHF outbreaks occurred sporadically during 2018–2019, both within and outside 'cattle corridor' districts of Uganda. Most cases were associated with tick exposure. The Ministry of Health should partner with the Ministry of Agriculture, Animal Industry and Fisheries to develop joint nationwide tick control programs and strategies with shared responsibilities through a One Health approach.

### Author summary

Uganda has had multiple Crimean-Congo haemorrhagic fever outbreaks since 2013 when the first outbreak was confirmed. Tick exposure has been identified as the major risk factor by our study and this finding was similar with other studies done during outbreaks in Uganda. However, Uganda still lacks national tick control guidelines and indiscriminate use of acaricides (pesticides specially for ticks) has been observed widely. This has been cited to influence increased tick resistance to acaricides. Our study might not indicate whether tick resistance to acaricides has increased tick populations in Uganda however it is imperative that tick control is considered in efforts of prevention and control of CCHF outbreaks. We therefore recommend improved tick control in Uganda through national regulations on acaricide distribution and use, development of strategies to reduce tick resistance to acaricides in the country, and more community-based engagement of tick control in livestock management.

## Introduction

Crimean-Congo haemorrhagic fever (CCHF) is a viral haemorrhagic fever (VHF) caused by a virus from the family *Bunyaviridae* genus *Nairovirus*. The disease is endemic in Africa and is one of the notifiable zoonotic diseases in Uganda [1]. Ticks that feed on infected livestock and subsequently feed on humans can transmit the virus between species, but contact with infectious blood or bodily secretions may also transmit the virus between livestock and humans, or from one human to another [2]. Various domestic livestock and wildlife act as amplifying hosts for the virus, but unlike humans, infected livestock do not exhibit clinical symptoms [3,4]. Globally, Crimean-Congo haemorrhagic fever virus (CCHFv) is found in Africa, Asia, and Europe; within these areas, shrub and grassland cover, which is favourable for ticks, is an important predictor of virus distribution [5]. The geographic distribution of the virus correlates with that of ticks belonging to the genus Hyalomma, which are the principal vectors. However, Rhipicephalus and Dermacentor ticks are also able to carry CCHFv hence can potentially be transmitted [6]. In a study in Iran, 4.6% infection rate was found in *Hyalomma* genus and 0.7% in *Rhipicephalus sanguineus* species; and both thrive in Uganda [7].

The incubation period for CCHFv in humans mainly depends on viral dose and route of infection [8,9]. Incubation is typically 1–9 days following a tick bite, and 5–13 days following contact with infectious fluids [1]. Humans infected with CCHFv can transmit the virus to other persons via blood or other bodily secretions following symptom onset [10]. Symptoms include sudden high-grade fever, headache, vomiting, back pain, joint pain, and stomach pain. Haemorrhaging may occur in later stages of the illness [1]. Between 10% and 40% of patients die of the disease. Real-time polymerase chain reaction (RT-PCR) is used for disease confirmation in the acute phase, whereas serological tests are used in the convalescent phase [4]. Case management is primarily through supportive treatment, although ribavirin treatment may be beneficial [4].

Uganda has had a VHF surveillance system since 2010. The program has more than 20 sentinel surveillance sites and receives samples for testing from all over Uganda as well as neighbouring countries, such as South Sudan and Rwanda. This increased surveillance has led to multiple confirmations of CCHF outbreaks–categorized as one or more cases of CCHF–in the intervening years [11]. During 2013–2017, Uganda documented eight confirmed CCHF outbreaks, all within the 'cattle corridor' but in geographically distinct districts. The 'cattle corridor' illustrated in Fig 1 is an area stretching from southwestern to northeastern Uganda and is dominated by pastoral rangelands. The first CCHF outbreak documented in the VHF surveillance system occurred in 2013 in Agago District, with three case-patients. During the next five years, fewer than 10 cases were reported in multiple independent outbreaks in Wakiso, Nakaseke, Kiboga, Luweero, and Mubende Districts. Notes from the field published after a CCHF outbreak in central Uganda found exposure to ticks as a risk factor for infection [12].

In July 2018, the Uganda Virus Research Institute (UVRI) (the national reference laboratory for VHF diagnosis in Uganda) reported two PCR-confirmed CCHF cases from Isingiro

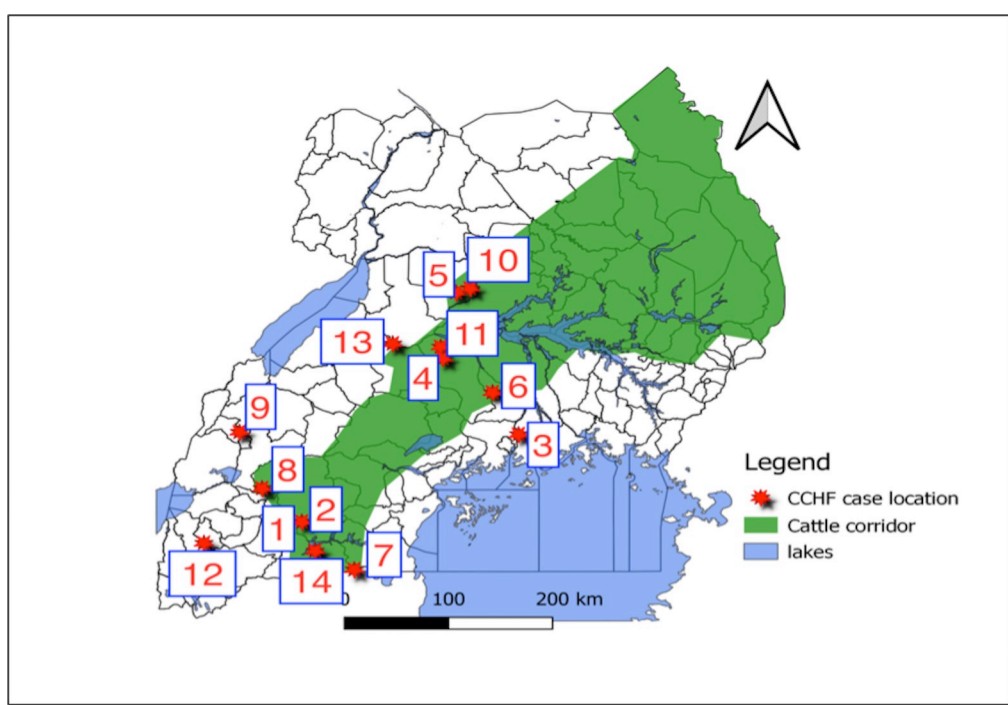

**Fig 1. Distribution of Crimean-Congo haemorrhagic fever case-patients confirmed between July 2018 and January 2019, highlighting the 'cattle corridor' and the sequence of symptom onset of the case-patients.**

District. Over the next seven months, nearly a dozen CCHF cases were confirmed in various districts, raising concern at the Ministry of Health about reasons for possible increase in frequency and geographical distribution of the disease. In July 2018, we began an investigation to determine the scope of the outbreaks during mid-2018 to early-2019, and to identify risk factors for disease.

## Methods

### Ethics statement

This investigation was in response to public health emergencies i.e. CCHF outbreaks and was therefore determined to be non-research. The Ministry of Health Uganda through the Office of the Director General of Health Services gave the directive and approval to investigate this outbreak. The Office of the Associate Director for Science, Centers for Global Health, CDC also determined that this activity was not human subject research, and its primary intent was public health practice or a disease control activity (specifically, epidemic or endemic disease control activity). We obtained verbal informed consent from case-patients during this investigation and other interviewed community members that were above 18 years. For participants below 18 years of age, we sought verbal consent from their parents or guardians and assent from the minors. We ensured confidentiality by conducting interviews in privacy ensuring that no one could follow proceedings of the interview. The questionnaires were kept under lock and key to avoid disclosure of personal information of the respondents to members who were not part of the investigation.

### Case definition and case-finding

We defined a suspected case as sudden onset of fever ($\geq 37.5°C$) with $\geq 4$ of the following signs and symptoms: anorexia, vomiting, diarrhoea, headache, abdominal pain, joint pain, or sudden onset of unexplained bleeding between 1 July 2018–30 January 2019 in a resident of one of the 11 affected districts. A confirmed case-patient was a suspected case-patient who tested positive for CCHFv by RT-PCR assay at UVRI. All laboratory procedures used in this laboratory for VHF diagnosis, including for CCHFv, have been previously published [13].

### Descriptive epidemiology and hypothesis generation

Due to the nonspecific symptoms in early infection, ill persons are typically not formally investigated for CCHF without a positive test. Thus, following case confirmation, investigation teams were dispatched to the field to obtain further details. We conducted descriptive epidemiology to collect information on demographics, clinical characteristics, location, and timing of infection and associated potential exposures. We computed attack rates using the projected population estimates, based on the 2014 census for the 11 affected districts [14]. We used QGIS to draw maps using Uganda Bureau of Statistics (UBOS) shapefiles [15].

To identify potential risk factors, we interviewed eight case-patients using a modified standard WHO VHF case investigation form [16]. We interviewed the case-patients about their potential exposures between the minimum and maximum incubation periods for CCHF (effective exposure period of 1–14 days).

### Case-control study

We conducted a case-control study that included 14 confirmed case-patients and four control-persons for each case-patient, matched by age (±10 years of case-patient age; for children <10 years, matched controls were +/-2 years of the case-patient age), sex, and sub-county. All

control-persons were asymptomatic during the 14 days before the case-patient's symptom onset. We collected data from both case-patients and control-persons on potential exposures during the effective exposure period (the two weeks before the case-patients' onsets of fever) using a structured questionnaire. Exposures considered included contact with livestock, identification of ticks on the body, consumption of raw milk, and direct contact with body fluids of a human or animal.

We calculated attack rates using population statistics of the districts from Uganda Bureau of Statistics and significance was calculated using chi-square tests. We computed crude odds ratios using simple 2x2 analysis in Epi Info 7. To account for the matched case-control study design, we stratified the data by the case-control sets and computed the Mantel-Haenszel odds ratios (OR) and 95% confidence intervals (CI) in Epi Info 7. Exact conditional logistic regression was used with variables that failed to converge during stratification.

## Results

### Descriptive epidemiology of confirmed CCHF case-patients

Four hundred and nine suspected case-patients were identified in Uganda during July 2018 to January 2019 through the VHF surveillance system. Of these, 14 case-patients were confirmed CCHFv-positive in 13 separate outbreaks. Two case-patients were confirmed in each of Nakaseke, Isingiro and Kiryandongo districts. One case-patient was confirmed in each of Mukono, Ibanda, Rakai, Luweero, Masindi, Rukungiri, Kiruhura and Kabarole districts. Of the 11 affected districts, eight (73%) were in the 'cattle corridor' (Fig 1). Most of the case-patients (57%) lived in savannah grasslands, while 29% lived in highlands.

Five case-patients died (case-fatality rate = 36%). The most frequent symptoms were fever (100%), bleeding (93%), general body weakness (79%), and abdominal pain (71%).

The mean age of case-patients was 24 years (range: 6–36) and the most affected age group was 20–44 years (9/1,255,997; attack rate (AR) = 7.2/1,000,000) compared to persons ≤19 years (5/2,474,438; AR = 2.0/1,000,000). The difference was statistically significant with p = 0.015. The AR was nearly twice as high in males (10/2,063,605; AR = 4.4/1,000,000) as in females (5/2,109,141; AR = 2.4/1,000,000), although this difference was not statistically significant (p = 0.27). More than half (57%) of the case-patients were herdsmen; four (29%) had no reported interaction with livestock.

The first of the 14 case-patients had onset on 7 July 2018 in Isingiro District. Case-patient illnesses occurred subsequently in other districts with no identifiable pattern in time and place. The last case-patient in this investigation had illness onset on 16 January 2019 in Kiruhura District (Table 1 and Fig 1).

The first two case-patients were a married couple from Isingiro District. The woman (wife) was the first case, she did not report a known tick bite, but lived near a kraal with tick-infested cattle. The second case-patient (husband) similarly did not report a tick bite but was exposed to the first case-patient (his wife) during his effective incubation period and had illness onset five days after his wife. This case-patient cared for his wife (who had haemorrhagic symptoms) during her illness and stayed with her in the hospital; his symptoms were mild (low-grade fever, general body weakness, sore throat and chest pain). Of the remaining cases, nine case-patients reported known tick bites (having found a tick on their bodies) during the 14 days before illness onset (Table 1) and three case-patients did not report tick bites (case-patients 3, 11, and 12), one lived in proximity to livestock infested with ticks at the time of our investigation, and one lived near goats, although they were not found to have ticks at the time of the study. However, one case-patient, from Mukono District, lived in a semi-urban area with no known contact with livestock. This case-patient visited a mosque where livestock were brought

**Table 1. Exposure information for case-patients with confirmed Crimean-Congo haemorrhagic fever during July 2018 to January 2019 in Western and Central Uganda.**

| Case-patient | District | Date of symptom onset | Reported tick bite during 14 days before illness | Suspected relevant exposure |
|---|---|---|---|---|
| 1 | Isingiro | 7 July 2018 | No | • Lived near a kraal<br>• Tick-infested cattle recently relocated for pasture |
| 2 | Isingiro | 12 July 2018 | No | • Likely infected by his wife, Case-Patient 1<br>• Lived near a kraal with tick-infested cattle |
| 3 | Mukono | 18 August 2018 | No | • Lived in a semi-urban area<br>• No known contact with livestock<br>• Visited a mosque towards Eid al-Adha when livestock for slaughter had been brought<br>• Suspected to have been exposed during this time |
| 4 | Nakaseke | 3 September 2018 | Yes | • Lived on a farm with livestock<br>• Livestock were not tick infested during the investigation<br>• Reported interaction with ticks, including biting ticks to kill them |
| 5 | Kiryandongo | 15 September 2018 | Yes | • Lived with two tick-infested oxen (hard ticks seen) |
| 6 | Luweero | 17 September 2018 | Yes | • Lived in an urban setting; owned tick-infested calf (hard ticks seen)<br>• Used to herd the calf |
| 7 | Rakai | 17 September 2018 | Yes | • Household next to grazing field<br>• Used to sleep on the floor in a temporary house constructed from tarpaulin and twigs<br>• Reported tick infestation in beddings |
| 8 | Ibanda | 3 October 2018 | Yes | • Slept in a semi-detached house with a room sharing a wall with a room that housed goats<br>• Reported that the church farm had received two tick-infested cattle in the two weeks before his onset |
| 9 | Kabarole | 15 October 2018 | Yes | • The goats had no ticks at the time of our investigation<br>• Household was close to a grazing ground shared with distant neighbours<br>• Reported that she 'always plucked ticks off her skin' every morning |
| 10 | Nakaseke | 20 October 2018 | No | • A child who lived on a farm with tick-infested livestock |
| 11 | Kiryandongo | 23 October 2018 | Yes | • Lived in a household with a kraal for tick-infested cattle<br>• Reportedly interacted with the cattle and removed ticks on his body daily |
| 12 | Rukungiri | 7 November 2018 | No | • Lived in highlands with goats<br>• No ticks identified on goats at time of visit<br>• Exposure still unclear |
| 13 | Masindi | 26 December 2018 | Yes | • Visited his father's farm before his onset<br>• Father reported the farm had tick-infested cattle |
| 14 | Kiruhura | 16 January 2019 | Yes | • Lived on a property with >100 head of tick-infected cattle<br>• Reportedly had no interaction with livestock |

for Eid al-Adha (Festival of Sacrifice in Islam) two days before symptom onset, which might have provided a source of exposure. Unfortunately, the patient died, and further details of her exposure were unavailable.

From the descriptive analysis and environmental assessment, we hypothesised that having livestock nearby during the exposure period, living within a 1-km radius of grazing fields, having livestock within the household compound and having tick-infested livestock within the household compound were risk factors.

### Risk factors for infection with CCHF

In the case-control study, we found that people living within 1 km of grazing fields for livestock were 18 times more likely to be CCHF case-patients than controls ($OR_{M-H}$ = 18, 95%CI = (3.2-∞); n = 12 (86%) cases and n = 27 (48%) controls lived < 1km) and people that identified ticks attached to their bodies during the effective exposure period were 9 times more likely

**Table 2. Association between risk factors and Crimean-Congo haemorrhagic fever infection during July 2018 to January 2019 in Western and Central Uganda.**

| Risk factor during 14 days before illness onset | Cases (%) | Controls (%) | Crude OR (95%CI) | OR$_{M-H}$ (95%CI) |
|---|---|---|---|---|
| Household had grazing fields within 1km | | | | |
| Yes | 12 (86) | 27 (48) | **6.4 (1.3–31)** | **18 (3.2-∞)*** |
| No | 2 (14) | 29 (52) | Ref. | Ref. |
| Noticed a tick attached to their body | | | | |
| Yes | 10 (71) | 15 (27) | **6.8 (1.9–25)** | **9.3 (1.9–46)** |
| No | 4 (29) | 41 (73) | Ref. | Ref. |
| Had livestock within household | | | | |
| Yes | 10 (71) | 31 (55) | 2.0 (0.56–7.2) | 2.5 (0.6–10) |
| No | 4 (29) | 25 (45) | Ref. | Ref. |
| Livestock in household had ticks | | | | |
| Yes | 7 (70) | 16 (50) | 2.3 (0.51–10) | 2.0 (0.35–11) |
| No | 3 (30) | 16 (50) | Ref. | Ref. |
| Had livestock in neighbourhood | | | | |
| Yes | 13 (93) | 36 (64) | 7.2 (0.88–59) | n/a |
| No | 1 (7) | 20 (36) | Ref. | |
| Neighbourhood livestock had ticks | | | | |
| Yes | 8 (80) | 13 (48) | 4.3 (0.77–24) | 8.3 (0.78–89) |
| No | 2 (20) | 14 (52) | Ref. | Ref. |
| Took ticks off livestock | | | | |
| Yes | 5 (36) | 10 (18) | 2.6 (0.7–9.3) | 2.7 (0.73–9.7) |
| No | 9 (64) | 46 (82) | Ref. | Ref. |
| Interacted with livestock† | | | | |
| Yes | 10 (71) | 27 (48) | 2.7 (0.75–9.6) | 4.3 (0.86–21) |
| No | 4 (29) | 29 (52) | Ref. | Ref. |
| Introduced new livestock to household | | | | |
| Yes | 4 (40) | 6 (19) | 2.9 (0.62–13) | 5.3 (0.73–38) |
| No | 6 (60) | 26 (81) | Ref. | Ref. |
| Drank milk | | | | |
| Yes | 9 (64) | 40 (71) | 0.72 (0.21–2.5) | 0.67 (0.17–2.7) |
| No | 5 (36) | 16 (29) | Ref. | Ref. |
| Drank raw milk | | | | |
| Yes | 5 (36) | 12 (21) | 2.0 (0.57–7.2) | 5.0 (0.64–39) |
| No | 9 (64) | 44 (79) | Ref. | Ref. |
| Drank macunde (fermented milk) | | | | |
| Yes | 4 (29) | 14 (25) | 1.2 (0.32–4.4) | 1.3 (0.25–7.1) |
| No | 10 (71) | 42 (75) | Ref. | Ref. |

*analysed using exact conditional logistic regression

† interacting with livestock involved herding, selling livestock, slaughtering, skinning and butchering carcasses, milking livestock or use of oxen to plough fields

to be CCHF case-patients than controls (OR$_{M-H}$ = 9.3, 95%CI = 1.9–46; n = 10 (71%) cases and n = 15 (27%) identified ticks attached to their bodies) shown in Table 2.

## Discussion

Human CCHF outbreaks in Uganda from mid-2018 to early 2019 were primarily associated with tick bites. This is a well-known risk factor and the findings are similar to those from previous CCHF outbreak investigations in Uganda [12,13] and globally [17–20]. A single case-patient in our investigation likely contracted the illness from another case-patient; his symptoms were mild, which is known to be associated with person-to-person routes of exposure for CCHFv [21,22], and his degree of exposure to the case-patient and the timing of his illness onset make person-to-person transmission likely.

Reports of CCHF outbreaks in Uganda are becoming more frequent. From 2013 to 2017, eight outbreaks were confirmed. An additional two outbreaks (not described in this

manuscript) occurred in early 2018. However, within just seven months during July 2018 to January 2019, 13 sporadic outbreaks were reported. It is unclear whether or not this is due to improved surveillance during this time period or a true increase in cases. In addition, an Ebola outbreak in the neighbouring Democratic Republic of Congo (DRC) was occurring during the same time period as this outbreak investigation [23], with the Ebola-affected area approximately 350km away from the Ugandan border of Mpondwe [23,24], and it is possible that the resulting heightened surveillance for VHF led to increased testing and diagnosis.

Currently, Uganda does not have a joint nationwide tick-control intervention or strategy despite past studies citing tick exposure as a risk factor for CCHF [25]. The choice to implement tick control measures in livestock is made individually, and at the cost of the livestock owner. However, individual tick control efforts will not be useful unless they are widely implemented. Having a grazing field near home was a strong independent risk factor for infection, likely because tick-infested livestock will sustain tick presence in grazing fields, even when one's own livestock do not harbour ticks. Communal or even regional concerted tick control efforts may be warranted to fully control this issue.

Beyond the issues with heterogeneous use of tick control across communities, the lack of national tick control guidelines has resulted in indiscriminate use of acaricides (pesticides specially for ticks), which has led to increased resistance among ticks in Uganda [26]. There have been recent documented increases in tick resistance to acaricides, which may have led to an increasing tick population. A study on tick resistance to acaricides in Western and Central Uganda found that 13% of ticks tested were resistant to the common acaricide Amitraz (mostly in Western Uganda), 43% to organophosphate-synthetic pyrethroid co-formulation, and 90% to selected synthetic pyrethroids. No ticks died when exposed to synthetic pyrethroids, and approximately 60% had super-resistance. These super-resistant ticks were found in abattoirs in Gulu located in Northern Uganda, showing the geographic spread of resistance around the country [26]. Increases in resistance could lead to greater tick infestations on livestock and subsequent increases in human CCHFv infections; however, this remains speculative since these studies do not necessarily focus on Hyalomma ticks and further investigation would be needed to evaluate this.

Our investigations established CCHF outbreaks in 11 districts. Although the districts are geographically distinct from each other, most are located within the "cattle corridor" [27]. In the cattle corridor, livestock are raised and are frequently traded within Uganda. Movement and trade of infected livestock has been found to modify geographical areas at risk for CCHF outbreaks [5]. In our study, the case-patient from Mukono District who lived in an urban setting–normally without risk of tick exposure–may have been exposed to a tick on livestock that had been transported to the urban mosque. Livestock trading is not strictly controlled in Uganda in terms of inspection and issuing of trade permits; livestock are often transported across the country in trucks indiscriminately. This kind of movement can easily introduce CCHFv to new territories [28,29].

Preventing CCHF outbreaks is challenging due to absence of an effective vaccine, widespread presence of ticks, and the fact that CCHFv infection is asymptomatic in livestock. Risk for exposure is always present in communities where residents have close contact with livestock. For this reason, prevention efforts should be strengthened through awareness and education efforts with regards to reducing transmission risk and improving tick control in such communities.

## Study limitations

A small number of persons [14] met the case definition however we matched each with four controls to add power to the study. We have no further information on tick species or whether

livestock were infected because we could not perform any analysis on blood and tick samples collected from livestock within households due to resource constraints.

## Conclusions, public health actions and recommendations

CCHF outbreaks in Uganda during 2018 and 2019 were sporadic, dispersed in multiple districts, and primarily associated with proximity to grazing fields and tick exposure. We educated affected communities on risk factors and prevention strategies for CCHFv transmission. The district rapid response teams were activated, including establishment of an emergency hotline for case reporting. Healthcare workers were trained in patient management, and infection control and isolation units were designated in general hospitals in the affected districts. No further cases have been confirmed in the villages that were sensitized as of August 2019.

CCHF is a zoonotic disease that affects humans however the infection does not cause disease in livestock. Nonetheless, there are other tick-borne infections that affect animal health. In Uganda, animal health is monitored under Ministry of Agriculture, Animal Industry, and Fisheries (MAAIF) while acaricides are regulated by National Drug Authority (NDA), an autonomous authority under Ministry of Health. We recommended that the Ministry of Health partners with MAAIF to develop joint nationwide tick control programs and strategies with shared responsibilities among the ministries through a One Health approach. We recommended that NDA improves its involvement in tick control strategies in Uganda by revising national regulations on acaricide distribution and use; developing strategies to reduce tick resistance to acaricides in the country through evidence-based use of acaricides supported by research [25]. More research studies should be conducted to identify acaricides to which ticks have developed resistance and the extent and geographical distribution that such ticks cover. We also recommended that a more community-based approach is used in tick control where community leaders mobilize the community towards communal tick-control efforts such as communal spraying livestock and rotation of acaricides used.

## Acknowledgments

We acknowledge and appreciate the contributions of Mbarara Regional Referral Hospital and Nakivaale Health Center III towards investigation and response during the multiple CCHF outbreaks. We also acknowledge participation of the District Health Teams of the affected districts during the investigation. We also appreciate Uganda Virus Research Institute for support in sample transportation and prompt diagnosis of CCHF.

## Author Contributions

**Conceptualization:** Bernadette Basuta Mirembe, Angella Musewa, Daniel Kadobera, Esther Kisaakye, Doreen Birungi, Benon Kwesiga, Steven Ndugwa Kabwama, Bao-Ping Zhu, Alex Riolexus Ario.

**Data curation:** Bernadette Basuta Mirembe, Angella Musewa, Daniel Kadobera, Esther Kisaakye, Doreen Birungi, Jimmy Baluku, Alex Riolexus Ario.

**Formal analysis:** Bernadette Basuta Mirembe, Angella Musewa, Daniel Kadobera, Doreen Birungi, Jimmy Baluku, Steven Ndugwa Kabwama, Bao-Ping Zhu, Julie R. Harris, Alex Riolexus Ario.

**Funding acquisition:** Alex Riolexus Ario.

**Investigation:** Bernadette Basuta Mirembe, Angella Musewa, Daniel Kadobera, Esther Kisaakye, Doreen Birungi, Stephen Balinandi, Alex Tumusiime, Jackson Kyondo, Sophia Mbula Mulei, Jimmy Baluku, Benon Kwesiga, Steven Ndugwa Kabwama, Alex Riolexus Ario.

**Methodology:** Bernadette Basuta Mirembe, Angella Musewa, Daniel Kadobera, Esther Kisaakye, Doreen Birungi, Daniel Eurien, Luke Nyakarahuka, Stephen Balinandi, Alex Tumusiime, Jackson Kyondo, Sophia Mbula Mulei, Jimmy Baluku, Benon Kwesiga, Steven Ndugwa Kabwama, Bao-Ping Zhu, Alex Riolexus Ario.

**Project administration:** Bernadette Basuta Mirembe, Daniel Kadobera, Alex Riolexus Ario.

**Resources:** Bernadette Basuta Mirembe, Julius Julian Lutwama, Alex Riolexus Ario.

**Software:** Bernadette Basuta Mirembe, Daniel Eurien, Julie R. Harris.

**Supervision:** Bernadette Basuta Mirembe, Daniel Kadobera, Luke Nyakarahuka, Stephen Balinandi, Bao-Ping Zhu, Julius Julian Lutwama, Alex Riolexus Ario.

**Validation:** Bernadette Basuta Mirembe, Daniel Kadobera, Luke Nyakarahuka, Stephen Balinandi, Alex Tumusiime, Jackson Kyondo, Sophia Mbula Mulei, Bao-Ping Zhu, Julie R. Harris, Julius Julian Lutwama, Alex Riolexus Ario.

**Visualization:** Bernadette Basuta Mirembe, Daniel Kadobera, Daniel Eurien, Luke Nyakarahuka, Stephen Balinandi, Bao-Ping Zhu, Julie R. Harris, Julius Julian Lutwama, Alex Riolexus Ario.

**Writing – original draft:** Bernadette Basuta Mirembe, Angella Musewa, Daniel Kadobera, Esther Kisaakye, Doreen Birungi, Daniel Eurien, Steven Ndugwa Kabwama, Bao-Ping Zhu, Julie R. Harris, Alex Riolexus Ario.

**Writing – review & editing:** Bernadette Basuta Mirembe, Daniel Kadobera, Luke Nyakarahuka, Stephen Balinandi, Steven Ndugwa Kabwama, Bao-Ping Zhu, Julie R. Harris, Alex Riolexus Ario.

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
