## [Decision Letter · Decision Letter 0]

11 Aug 2020

Dear MISS Mirembe,

Thank you very much for submitting your manuscript "Sporadic Outbreaks of Crimean-Congo Haemorrhagic Fever in Uganda, July 2018-January 2019" for consideration at PLOS Neglected Tropical Diseases. As with all papers reviewed by the journal, your manuscript was reviewed by members of the editorial board and by several independent reviewers. In light of the reviews (below this email), we would like to invite the resubmission of a significantly-revised version that takes into account the reviewers' comments. 

Dear Dr. Mirembe and co-authors, Thank you very much for your submission to PLoS NTD on CCHF in Uganda. Your manuscript was reviewed by two leading experts in the fields of epidemiology and CCHFV. Both reviewers felt that some re-examination of the interpretation of our findings was warranted before further consideration for publication. Please pay close attention to their helpful comments and suggestions. I look forward to receiving a revised manuscript at your earliest convenience. Please stay safe, healthy, and in good spirits during these challenging COVID times. Yours, Dr. Brian Bird - UCD Davis One Health Institute, Associate Editor PLoS NTD

We cannot make any decision about publication until we have seen the revised manuscript and your response to the reviewers' comments. Your revised manuscript is also likely to be sent to reviewers for further evaluation.

Sincerely,

Brian Bird, DVM, ScM, PhD

Associate Editor

Aysegul Taylan Ozkan

Deputy Editor

Dear Dr. Mirembe and co-authors, Thank you very much for your submission to PLoS NTD on CCHF in Uganda. Your manuscript was reviewed by two leading experts in the fields of epidemiology and CCHFV. Both reviewers felt that some re-examination of the interpretation of our findings was warranted before further consideration for publication. Please pay close attention to their helpful comments and suggestions. I look forward to receiving a revised manuscript at your earliest convenience. Please stay safe, healthy, and in good spirits during these challenging COVID times. Yours, Dr. Brian Bird - UCD Davis One Health Institute, Associate Editor PLoS NTD

Reviewer's Responses to Questions

**Key Review Criteria Required for Acceptance?**

**Methods**

-Are the objectives of the study clearly articulated with a clear testable hypothesis stated?

-Is the study design appropriate to address the stated objectives?

-Is the population clearly described and appropriate for the hypothesis being tested?

-Is the sample size sufficient to ensure adequate power to address the hypothesis being tested?

-Were correct statistical analysis used to support conclusions?

-Are there concerns about ethical or regulatory requirements being met?

Reviewer #1: The objectives are clear. The testable hypotheses could be better described. Is your goal to see if risk factors in Uganda are similar to known risk factors in other endemic countries? Or do do you want to know whether exposure was due to human contact, tick exposure or livestock exposure so that mitigation measures can be properly allocated? 

A matched case-control study design is appropriate for this study. 

Sample size: 14 confirmed cases is low, however given that a large geographic area was evaluated and a significant number of suspect cases (n=409) were explored, the methods were appropriate. I would like further information on how the 14 confirmed cases were identified. Were all 409 suspects tested and there were only 14 confirmed RT-PCR positives? How many medical facilities were included in this surveillance program? How were they distributed within districts? What kinds of facilites were they (referral hospitals, local medical clinics, etc.)? 

Calculating attack rates don’t seem appropriate with such small case numbers and likely many missed cases. I suggest analyzing your data regarding age as an odds ratio. 

I suggest, re-evaluated your stratification methods for calculation of your Mantel-Haenzel odds ratios so that each stratum has larger sample size numbers. Stratification into 14 categories (based on matched case-control subsets, isn’t going to give you large enough sample sizes for each strata). Try stratifying by age or sex and see if that significantly changes your adjusted values.

Reviewer #2: the methods are clearly described and appropriate for the study.

**Results**

-Does the analysis presented match the analysis plan?

-Are the results clearly and completely presented?

-Are the figures (Tables, Images) of sufficient quality for clarity?

Reviewer #1: Methods for logistic and exact logistic regression models are described in the methods but I don’t see any results of this described. 

I suggest the following sub-headings for results section: 1. Descriptive epidemiology of confirmed CCHF cases, 2. Risk factors for infection with CCHF

The description of results in the case-control portion of the study could be more clearly presented. I suggest presenting the significant risk factors first in a format that focuses on risk. For example, “The odds of living within 1 km of grazing fields for livestock were 18 times higher among CCHF case-patients than controls (ORMH = 18, CI=___; n = 12 (86%) cases and n = 27 (48%) controls lived < 1km).” 

Table 1: 5th column title could be “Potential routes of exposure”. This column could be presented in more bullet point form than paragraphs. That would consolidate this table. 

Was being within the cattle corridor a significant risk factor? I suggest adding this to your table.

Reviewer #2: The results are clearly presented and the tables are suitable for concise and clear presentation of the results.

**Conclusions**

-Are the conclusions supported by the data presented?

-Are the limitations of analysis clearly described?

-Do the authors discuss how these data can be helpful to advance our understanding of the topic under study?

-Is public health relevance addressed?

Reviewer #1: The conclusions are supported by the data presented. Limitations of the study could be further described. 

The recommendations surrounding tick control are helpful and relevant for this topic and are specific to this region. I’m curious why butchering / slaughtering / handling livestock carcasses and bodily fluids was not investigated as potential risk factors? There are distinct differences with regard to virus exposure between rearing livestock and the act of butchering and it would have been interesting to see if there was a distinction between these two activities in this region. Could this analysis be added? This would help public health authorities to prioritize interventions around tick exposure as opposed to safe handling of livestock carcasses. 

The public health relevance was thoroughly discussed.

Reviewer #2: Public health relevance is addressed in the study. 

My main concern with this manuscript is that the authors have indicated the need to implement tick control as a method to reduce the burden of CCHFV infections and make reference to possible increased tick resistance due to acaricide control. However although increased tick control would be recommended nowhere in their manuscript do they actually mention that CCHFV is transmitted by ticks belonging to the genus Hyalomma, which are considered to be the principal vectors. although the virus has been isolated from some 30+ species of tick, the role of these ticks as vectors is not clear. In addition the paper they refer to for increased tick resistance was based on data mostly for Rhipicephalus species (no Hyalomma were identified in the paper). hence although the indiscriminate use of acaracides may have influenced hyalomma populations there is no evidence from the manuscript cited to support this. this needs to be made clearer in the manuscript.

what tick species occur in the districts investigated? was there any information on the tick species described in this study? if no information then please include this as a limitation of the study.

Line 251: the case living in an urban area may also have been exposed to infected blood during slaughter of animals as religious festivals and so this sentence regarding tick exposure is not justified and considering there was evidence of tick-bite.

**Editorial and Data Presentation Modifications?**

Reviewer #1: Line 24 – 25 – Reword - >4 what? (days or symptoms?)

Line 27 – Check on distinction between using CCHF and CCHFV throughout. You test for the virus but diagnose the disease. 

Line 35 – 36 – was this difference in attack rates statistically significant? 

Line 46 – What is the One Health aspect of control you are recommending? Tick control for both the environment and animals? This could be clarified. 

Line 66 – “caused by a tick-born virus in the genus Nairovirus, family Orthobunyaviridae”

LINE 80 – “other bodily secretions”

Line 93 – reference figure 1 when describing the “cattle corridor”

Line 109 – “…occurring between 1 July 2018-30 in a resident of 1 of the 11 affected districts.” 

Line 118 – “We conducted a descriptive epidemiology investigation to collect information on symptoms, demographics, location and timing of infection and associated activities.”

Line 122 – “To identify potential risk factors for infection, we interviewed……”

Line 124 – 125 – What was the effective exposure period that you used? 

Line 136 – what do you mean by “population statics”?

Line 137 – “significance was calculated…”

Line 145 – Were the non-confirmed suspect case-patients negative on PCR or not tested? 

Line 190 – 193- If you’re going to state hypotheses or describe known risk factors for CCHF from other studies, this information would be better placed in the introduction. 

Line 223 -224. This last sentence is redundant.

Reviewer #2: Lines 6 to 7: Please correct, CCHF is caused by Crimean-Congo hemorrhaguc fever orthonairovirus a member of the Orthonairovirus genus and the family Nairoviridae 

Line 73: Crimean-Congo haemorrhagic fever orthonairovirus (CCHFV) is a tick-borne virus found in Africa, Asia, eastern Europe and the Balkans. 

I suggest adding : The geographic distribution of the virus correlates with that of ticks belonging to the genus Hyalomma, which are considered to be the principal vectors.

Line 76-77: the authors state that The incubation period for CCHF in humans depends on viral dose, genetic factors and immune status of the host, and route of infection (6, 7). it is accepted that the incubation period does vary with source of infection, that is tick bite versus exposure to infected tissues, but the role of viral dose and genetic factors is not so clearly defined. I am unsure what is referred to when saying immune status influences incubation period.

Line 78: authors state that Incubation is typically 1-9 days following a tick bite, and 5-13 days following contact with infectious fluids. it would be more accurate to indicate that incubation period is usually 2 to 3 days following tick bite although occasionally longer the incubation period is typically short after tick bite.

Please provide references for all instances where previous cases in Uganda are referred to.

**Summary and General Comments**

Reviewer #1: This was an important epidemiological investigation of CCHF cases in Uganda that explored several well-known risk factors for infection with the virus in endemic countries, in the specific context of Uganda. The finding that tick bites were the major risk factor for infection as opposed to direct contact with livestock has important implications for public health messaging and interventions in Uganda.

Reviewer #2: the authors have submitted a paper that will contribute to our knowledge of CCHF infections in Uganda and hence is an important paper but I suggest some of the limitations of the study especially with regard to the lack of information on tick species be considered.

PLOS authors have the option to publish the peer review history of their article (what does this mean?). If published, this will include your full peer review and any attached files.

Reviewer #1: No

Reviewer #2: No
---

## [Decision Letter · Decision Letter 1]

15 Dec 2020

Dear MISS Mirembe,

Thank you very much for submitting your manuscript "Sporadic Outbreaks of Crimean-Congo Haemorrhagic Fever in Uganda, July 2018-January 2019" for consideration at PLOS Neglected Tropical Diseases. As with all papers reviewed by the journal, your manuscript was reviewed by members of the editorial board and by several independent reviewers. The reviewers appreciated the attention to an important topic. Based on the reviews, we are likely to accept this manuscript for publication, providing that you modify the manuscript according to the review recommendations. 

Dear Authors, Thank you for the hard work on your revision. One of the reviewers has a few more minor comments that when addressed will further enhance the readability and clarity of your manuscript. We look forward to your revised and updated manuscript. Yours, -Brian Bird, PLoS NTD Associate Editor; UC Davis One Health Institute

Sincerely,

Brian Bird, DVM, ScM, PhD

Associate Editor

Aysegul Taylan Ozkan

Deputy Editor

Dear Authors, Thank you for the hard work on your revision. One of the reviewers has a few more minor comments that when addressed will further enhance the readability and clarity of your manuscript. We look forward to your revised and updated manuscript. Yours, -Brian Bird, PLoS NTD Associate Editor; UC Davis One Health Institute

Reviewer's Responses to Questions

**Key Review Criteria Required for Acceptance?**

**Methods**

-Are the objectives of the study clearly articulated with a clear testable hypothesis stated?

-Is the study design appropriate to address the stated objectives?

-Is the population clearly described and appropriate for the hypothesis being tested?

-Is the sample size sufficient to ensure adequate power to address the hypothesis being tested?

-Were correct statistical analysis used to support conclusions?

-Are there concerns about ethical or regulatory requirements being met?

Reviewer #1: Overall the methods were appropriate for the data and to address the stated objectives. There is a discrepancy between descriptions of the case-definition between the main text and the abstract. One states all cases were confirmed by PCR and the other says PCR or IgM serology. This is an important distinction that should be clarified. Including only cases that tested positive by PCR would be stronger. 

"Exact conditional logistic regression was used with variables that failed to converge during stratification." I'm not following this part. There shouldn't have been model convergence issues with calculating M-H Odds ratios. Is this description referring to your method for calculating the crude Odds ratios that you report in the results? If so, I would state it like that. 

The population is clearly described and appropriate for the hypothesis being tested. The number of confirmed cases is small but given the type of study, the analyses and the use of 4 matched controls per case was appropriate. There are no concerns about ethical or regulatory requirements being met.

Reviewer #2: The authors have addressed reviewer comments

**Results**

-Does the analysis presented match the analysis plan?

-Are the results clearly and completely presented?

-Are the figures (Tables, Images) of sufficient quality for clarity?

Reviewer #1: Table 2: The sample sizes for individual acaricides in the table is small and no statistics at the individual level are provided, so I would remove these. Frequency of spraying livestock also doesn't seem to fit as a "risk factor" for developing CCHF in this table because no analyses of them as risk factors are provided. For the "Butchering meat" and "Slaughtering animals" factors can those be categorized as something other than N/A's. Weighted correction methods such as the Haldane-Anscombe correction could be used to calculate odds ratios when one of the categories is 0. For some risk factors not all of the cases are represented (i.e. slaughtering animals only has data from 10 cases). Was data not collected for some of these cases on these topics? 

Study Limitations: I don't think you need to state these. You did a nice risk factor study regardless of whether your original objectives were met and this section feels a bit like it comes out of the blue because it wasn't mentioned earlier as an objective of the study.

Reviewer #2: the authors have addressed previous comments

**Conclusions**

-Are the conclusions supported by the data presented?

-Are the limitations of analysis clearly described?

-Do the authors discuss how these data can be helpful to advance our understanding of the topic under study?

-Is public health relevance addressed?

Reviewer #1: Conclusions are supported by the data presented. The conclusion section could take more of an active voice as you are making these recommendations to the people reading the paper in addition to the people you have already informed on the recommendations. The public health relevance is clearly addressed. 

The study limitations could mention the small number of cases that met the case definition. The discussion of development of resistance to acaricides amid inconsistent use is interesting and highly relevant to future public health interventions. Perhaps a short discussion of recommendations for future research in this area, to identify specific acaricides in which ticks have developed resistance to in Uganda is worth mentioning.

Reviewer #2: the authors have addressed previous comments

**Editorial and Data Presentation Modifications?**

Reviewer #1: Throughout: double check the use of CCHF vs CCHFV to make sure you are referring to the disease or the pathogen the way intended

Line 27: should be "CCHFV"

Line 37: missing an "=" sign for P value

Line 45: I like that you are calling out the One Health approach here but it's a bit vague what the context is. Are you meaning that preventive measures for tick control should be taken for both livestock and humans to lessen human exposure to tick bites? A sentence of clarification would help here. 

Line 70: "or from one human to another"

Line 71: the term "wild livestock" sounds contradictory; change to "domestic livestock and wild ungulates", or "wildlife"

Line 72: use CCHFV

Line 74: "important predictor of virus distribution"

Line 77: CCHFV

Line 80: CCHFV

Line 82: CCHFV 

Line 83: should be "other bodily secretions". 

Line 117: CCHFV

Line 119: CCHFV

Line 153: CCHFV-positive

Line 185: should be "did not report tick bites"

Table 1: Use periods at the end of each bullet or don't use them but be consistent throughout table.

Line 202: "In the case-control study, we found that people living within 1 km of grazing fields for livestock were 18 times more likely to be CCHF case-patients than controls"

Line 220: CCHFV

Line 261: change to "at risk for CCHF outbreaks". 

Line 266: can use CCHF virus or CCHFV but be consistent throughout paper

Line 268: Use alternative word than "interrupting", such as "preventing"

Line 276: "similar strains of CCHFV"

Reviewer #2: no comment

**Summary and General Comments**

Reviewer #1: This research identifies the most common risk factors associated with CCHF outbreaks in Uganda which has implications for implementation of public health measures to prevent this disease. It's valuable to know that tick bites as opposed to exposure to bodily fluids from livestock appears to be the major driver of cases in Uganda. The case information is clearly presented. Table 2 could use some further consideration on what should be presented as a risk factor as described above.

Reviewer #2: no additional comments

PLOS authors have the option to publish the peer review history of their article (what does this mean?). If published, this will include your full peer review and any attached files.

Reviewer #1: No

Reviewer #2: No
---

## [Editor Report · Decision Letter 2]

5 Feb 2021

Dear MISS Mirembe,

We are pleased to inform you that your manuscript 'Sporadic Outbreaks of Crimean-Congo Haemorrhagic Fever in Uganda, July 2018-January 2019' has been provisionally accepted for publication in PLOS Neglected Tropical Diseases.

Best regards,

Brian Bird, DVM, ScM, PhD

Associate Editor

Aysegul Taylan Ozkan

Deputy Editor

Dear Authors, Thank you for your submission to PLoS NTD and your subsequent hard work in responding to the reviewers comments and suggestions. I look forward to seeing your manuscript on-line to share your data on CCHF across Uganda. Yours, -Brian Bird; UC Davis One Health Institute, PLoS NTD Associate Editor

---

## [Editor Report · Acceptance letter]

1 Mar 2021

Dear MISS Mirembe,

We are delighted to inform you that your manuscript, "Sporadic Outbreaks of Crimean-Congo Haemorrhagic Fever in Uganda, July 2018-January 2019," has been formally accepted for publication in PLOS Neglected Tropical Diseases.

Best regards,

Shaden Kamhawi

co-Editor-in-Chief

Paul Brindley

co-Editor-in-Chief
